# Evidence from COVID-19 Patients and Murine Studies for a Continuing Trend Towards Targeting of Nasopharyngeal Ciliated Epithelial Cells by SARS-CoV-2 Omicron Sublineages

**DOI:** 10.3390/v17121631

**Published:** 2025-12-17

**Authors:** Agnes Carolin, Cameron R. Bishop, Kexin Yan, Branka Grubor-Bauk, Mark P. Plummer, Bing Tang, Michael Leitner, Eamon Raith, Simon C. Barry, Christopher M. Hope, Wilson Nguyen, Daniel J. Rawle, Andreas Suhrbier

**Affiliations:** 1Infection and Inflammation Department, QIMR Berghofer, Brisbane, QLD 4029, Australia; agnes.carolin@qimrb.edu.au (A.C.); cameron.bishop@qimrb.edu.au (C.R.B.);; 2Faculty of Health, Medicine and Behavioural Sciences, University of Queensland, St. Lucia, QLD 4072, Australia; 3Viral Immunology Group, Adelaide Medical School, Faculty of Health and Medical Sciences, University of Adelaide, Adelaide, SA 5005, Australia; 4Intensive Care Unit, Royal Adelaide Hospital, Adelaide, SA 5000, Australia; 5GVN Center of Excellence, Australian Infectious Disease Research Centre, Brisbane, QLD 4029 and 4072, Australia; 6School of Chemistry and Molecular Biosciences, University of Queensland, St. Lucia, QLD 4072, Australia

**Keywords:** SARS-CoV-2, omicron sublineages, COVID-19, nasopharyngeal swab, ciliated epithelia, RNA-Seq, mouse, nasal turbinates, immunohistochemistry

## Abstract

We describe RNA-Seq analyses conducted on nasopharyngeal swabs collected from 37 patients admitted to an Australian intensive care unit from October 2022 to August 2023. During this time, the dominant omicron sublineage infections broadly progressed from BA.5 to BA.2-like, to XBB-like, then XBC, consistent with global trends. Viral load and patient metadata correlations indicated this cohort was broadly representative of severe COVID-19 patients. Human gene expression analyses were complicated by the large range (>5 log) and variability in viral reads. Nevertheless, the comparison of XBC and BA.5 samples that had comparable viral read counts, revealed differentially expressed genes and a cellular deconvolution signature that indicated increased targeting of ciliated epithelial cells by XBC. To obtain more evidence for increased targeting of ciliated epithelial cells by the later omicron sublineage viruses, a series of mouse strains were infected with a BA.5 or a XBB isolate. Increased infection of the nasal turbinates and ciliated epithelial cells by XBB was demonstrated by viral titrations and immunohistochemistry, respectively. Compared with previous lineages, the omicron lineage showed increased targeting of ciliated epithelia in the upper respiratory tract, with the data presented herein suggesting this trend continued for the omicron sublineages.

## 1. Introduction

SARS-CoV-2, the virus that causes COVID-19, has evolved substantially since its emergence in late 2019 [1], with the omicron lineage emerging in 2022 and markedly diversifying thereafter into a series of sublineages [2,3,4,5]. Ongoing mutations in the spike protein are central to the classification of sublineages and are associated with increased transmissibility, immune evasion, and receptor binding [6,7]. However, other SARS-CoV-2 genes [8] are also undergoing evolutionary changes [9] that impact a range of virus-host interactions [10,11,12]. Omicron lineage viruses were characterized by increased replication in the upper (vs. the lower) respiratory tract [12] and a great capacity to infect ciliated epithelial cells [13,14,15] when compared with earlier lineage viruses. Infection of ciliated epithelial cells has several advantages for the virus including disruption of mucocilary clearance, a key defense against infection by, and spread of, SARS-CoV-2 [16,17,18].

SARS-CoV-2 infection is generally diagnosed by RT-PCR or protein-based diagnostic tests performed on material collected from patients via nasopharyngeal swabs [19,20]. Such sampling collects not only viral RNA and protein, but also host cells and cellular debris from the nasopharyngeal tract. Such swabs thus provide a readily available source of both viral RNA and human cellular mRNA from a major site of infection in COVID-19 patients [21]. Herein, we undertook RNA-Seq analysis of nasopharyngeal swabs collected from 37 patients admitted to the ICU of the Royal Adelaide Hospital, Australia from October 2022 to August 2023. Omicron sublineage viruses were identified in 33 of these patient samples, with RNA-Seq illustrating a >5 log range in viral reads per million (RPM). Correlations between patient metadata and viral RPM supported and were generally consistent with, previously published studies, indicating that this cohort was broadly representative of severe COVID-19 patients. Importantly, RNA-Seq also provided information on human gene expression in the nasopharynges of patients. Although complicated by the large range in viral RPM, bioinformatic analyses suggested increased targeting of ciliated epithelial cells by the later emerging omicron sublineage, XBC, when compared with the earlier sublineage, BA.5-like viruses.

We have previously described isolation of omicron virus isolates, BA.5 in 2022 and XBB in 2023, from nasopharyngeal swabs donated to research laboratories by (non-hospitalized) consented COVID-19 patients in Brisbane, Australia [22,23,24]. These two isolates were used to infect a series of mouse strains. Using virus titrations, RNA-Seq and immunohistochemistry (IHC), significantly higher levels of infection, including increased infections of ciliated epithelial cells, were seen in the nasal turbinates of mice infected with the later omicron sublineage XBB isolate, when compared with the earlier omicron sublineage BA.5 isolate.

The data presented herein provides evidence from a human cohort and murine experiments, which suggest that the evolutionary trend for increased targeting of ciliated epithelial cells by the omicron lineage [13,14,15], continued for the omicron sublineages.

## 2. Materials and Methods

### 2.1. Ethics and Regulatory Compliance

Human: Informed consent was obtained from all patients involved in the study. Nasopharyngeal swabs were collected from 49 COVID-19 patients admitted to the ICU of the Royal Adelaide Hospital from October 2022 to August 2023; (out of 49 samples, 37 yielded sufficient RNA for sequencing). The consent form (approved 29 April 2020) is provided in Appendix A. Swabs were placed into TRIzol^®^ reagent (Invitrogen) and stored at −80 °C prior to transport to QIMR Berghofer. Approval was obtained from the Central Adelaide Local Health Network (CALHN) Human Research Ethics Committee (CALHN13050; 29 April 2020). Approval was also obtained from the QIMR Berghofer Human Research Ethics Committee (Ref: P3600; 16 May 2024), with all samples and patient data deidentified prior to being sent to QIMR Berghofer. Research at QIMR Berghofer was approved by the Institutional Safety Committee.

The ethical and consent processes associated with isolation of SARS-CoV-2 virus isolates from nasopharyngeal swabs donated by COVID-19 patients in Brisbane has been previously described in detail [25].

Animal: All mouse work was conducted in accordance with the Australian code for the care and use of animals for scientific purposes (National Health and Medical Research Council, Australia). Mouse work was approved by the QIMR Berghofer Animal Ethics Committee (P3600; A2108-612). All infectious SARS-CoV-2 work was conducted in the BioSafety Level 3 (PC3) facility at the QIMR Berghofer MRI (Department of Agriculture, Fisheries and Forestry, certification Q2326 and Office of the Gene Technology Regulator certification 3445). Breeding and use of GM mice was approved under a Notifiable Low Risk Dealing (NLRD) Identifier: NLRD_Suhrbier_Oct2020: NLRD 1.1(a). Mice were euthanized using carbon dioxide.

### 2.2. RNA-Seq and Bioinformatics of Human Nasopharyngeal Samples

RNA was extracted from TRIzol according to manufacturer’s instructions. DNAse I treatment (New England Biolabs, Notting Hill, VIC 3168, Australia) and RNA clean-up was undertaken using RNeasy MinElute Cleanup Kit (QIAGEN, Clayton, VIC, Australia), according to the manufacturer’s instructions. The Agilent 4200 Tape Station was used to provide RNA Integrity Number (RIN) scores, a measure of RNA integrity [26]. Library preparations and ribosomal RNA depletions were undertaken using TrueSeq Stranded Total RNA Sample Preparation Kit, with Ribo-Zero (Illumina, Melbourne, VIC 3000, Australia). TapeStation D1000 Screen Tape (Agilent Technologies, Mulgrave, VIC 3170, Australia) was used to assess the fragment size of the completed libraries; samples where cDNA amplicons could not be detected did not progress to sequencing. ThermoFisher Qubit 4 Fluorometer (Thermofisher, Woolloongabba, QLD 4102, Australia was used to measure the concentration of the completed libraries. Sequencing was performed using NextSeq 2000 using the flow cell P3 Reagent (200 cycles), providing 76 bp paired-end reads. All samples were run in a single sequencing run; Q30 was 89.97%.

STAR aligner was used to align the RNA-Seq reads to a combined SARS-CoV-2 BA.5 (GenBank: OP604184.1) and human (GRCh38, version 38) reference genome [22]. Samtools v1.16 [27] was used to generate viral read counts expressed as viral reads per million reads (RPM) as described [28]. RPM refers to primary proper read pairs per million total paired end aligned reads obtained for each sample. RSEM v1.3.1 was used to generate read counts for host genes; differentially expressed genes (DEGs) were identified using DESeq2 v1.40.2 [29]. DEGs were analyzed using Ingenuity Pathway Analysis (IPA, v84978992) (QIAGEN). Relative abundance of specific cell types was estimated via cellular deconvolution using the SpatialDecon package in R [23,30], with cell-type expression matrices obtained from the human lung cell atlas [31] (https://data.humancellatlas.org/explore/projects/c4077b3c-5c98-4d26-a614-246d12c2e5d7, accessed on 10 November 2025).

### 2.3. RT-qPCR of Human Nasopharyngeal Samples

The cDNA synthesis (from the purified RNA from the nasopharyngeal swab samples) was undertaken using ProtoScript II First Strand cDNA Synthesis Kit (New England Biolabs). RT-qPCR was performed using iTaq Universal Probes Supermix (Bio-Rad) and SARS-CoV-2 (Sarbeco-E) primers; Forward: 5′-ACAGGTACGTTAATAGTTAATAGCGT-3′, Reverse: 5′-ATATTGCAGCAGTACG CACACA-3′ [32]. The florescent probe was 5′-FAM-ACACTAGCCATCCTTACTGCGCTTCG-ZEN™/Iowa Black^®^ FQ-3′ (Integrated DNA Technologies, Boronia, VIC 3155, Australia). BioRad CFX96 was used to perform RT-qPCR with the following cycling conditions: incubation at 50 °C for 10 min, followed by 3 min at 95 °C, then 40 cycles of 15 s at 95 °C and 30 s at 60 °C.

### 2.4. Phylogenetic Tree Generation

The Integrative Genomics Viewer (IGV; v.2.9.4) [33] was used to align the viral sequences retrieved from RNA-Seq to an original SARS-CoV-2 strain isolate (GISAID Accession ID: EPI_ISL_407896 and GenBank, Accession ID: MW772455.1). Consensus spike sequences were extracted from IGV, followed by translation via Expasy (online tool: https://web.expasy.org/translate/, accessed on 10 November 2025). MegaX (v.11) was used to generate the phylogenetic tree. The phylogenetic tree was constructed using the Maximum Likelihood method and General Time Reversible model and 1000 bootstrap replicates. Sublineages were classified further based on their spike mutations using GISAID.

### 2.5. BA.5 and XBB Virus Isolates and Growth Kinetics in Different Cell Lines

The XBB isolate (SARS-CoV-2_UQ01_) was obtained from nasopharyngeal aspirates of consented COVID-19 patient at the University of Queensland, Brisbane, Australia using Vero E6-TMPRSS2 cells as described [22,24]. The isolate (deposited as hCoV-19/Australia/UQ01/2023; GISAID EPI_ISL_17784860) is XBB.1.9.2.1.4 (Pango EG.1.4). The omicron BA.5 isolate, SARS-CoV-2_QIMR03_ (SARS-CoV-2/human/AUS/QIMR03/2022) belongs to the BE.1 sublineage (GenBank: OP604184.1) and was isolated at QIMR Berghofer from nasal swabs from consented COVID-19 patients as described previously [22,23]. Virus stocks were propagated in Vero cells, and titered using cell culture infectious dose 50% (CCID_50_) assays on Vero cells as described [34].

Growth kinetics were undertaken using the following cell lines; Vero E6 (C1008) [35], Vero E6-TMPRSS2 (Vero E6 expressing TMPRSS2 via lentiviral transduction) [36], HEK293T-mACE2, and HEK293T-hACE2 (HEK293T cells expressing mouse and human angiotensin-converting enzyme 2, respectively, via lentiviral transduction) [34], and Caco-2 (human epithelial cell line from a colon carcinoma) (ATCC HTB-37) [35]. Cells were seeded at 2x10^5^ cells per well (12-well flat-bottom plates), a day before the infection. Cells were infected with XBB or BA.5 (MOI = 0.1) (4 replicate wells per virus) for 1 h, cells washed 3× with PBS, culture medium added, plates placed into a standard incubator (37 °C, 5% CO_2_), and supernatants collected at the indicated times. Viral titers in the supernatants were determined by CCID_50_ assays using Vero E6 cells [34].

### 2.6. Mouse Strains

C57BL/6J mice (females seven weeks old) were obtained from the Animal Resources Centre (Ozgene), Perth, WA, Australia. Homozygous K18-hACE2 mice (expressing the human angiotensin-converting enzyme 2 from a keratin 18 promoter) (males 7–22 weeks old) were generated and bred in-house at the QIMR Berghofer Animal Facility from heterozygous K18-hACE2 mice (Jackson Laboratory, Bar Harbor, ME, USA; B6.Cg-Tg(K18-ACE2) 2Prlmn/J, JAX Stock No: 034860) as described [22]. Type I interferon (IFN) receptor 1-deficient *Ifnar*-/- mice on a C57BL/6J background [37] (females 24–36 weeks old) were bred in-house at QIMR Berghofer, and were originally provided by Dr P. Hertzog (Monash University, Melbourne, Australia) [38]. For K18-hACE2 and *Ifnar*-/- mice, animals were selected so the mean age and age distribution were similar in each group. Adaptive B and T cell response deficient *Rag1*-/- mice on a C57BL/6 background (B6.129S7-Rag1tm1Mom/J; Jackson Laboratory) (females eight weeks old) were also bred in-house. Mice were kept in standard animal house conditions; for details see [35].

### 2.7. Mouse Infections with XBB and BA.5 Viruses, Virus Titrations, and Immunohistochemistry

Mice received intrapulmonary infections of XBB and BA.5 viruses delivered via the intranasal route with 5 × 10^4^ CCID_50_ of virus in 50 μL RPMI 1640 whilst under light anesthesia, as described [16,22]. Mice were monitored as described [16] and euthanized using CO_2_. Nasal turbinates and lungs were harvested at the indicated times, weighed, stored frozen at −80 °C and then homogenized in medium and virus titers per gram determined by CCID_50_ assays as described [35]. Immunohistochemistry was undertaken on formalin-fixed whole heads (decalcified and paraffin embedded) using purified anti-SARS-CoV-2 spike monoclonal antibody, SCV2-1E8, as described [24,39]. Slides were scanned using Aperio AT Turbo (Aperio, Vista, CA, USA) and images collected using Aperio ImageScope software v10 (Leica Biosystems, Mt Waverley, VIC 3149, Australia). Images (.svs) were examined by a trained European-board certified veterinary pathologist.

### 2.8. Statistics

Human data: Correlations were undertaken using SPSS Statistics (v23) (IBM, Armonk, NY, USA) and Fischer’s exact tests were performed using JNP Pro 18 (SAS Institute, Cary, NC, USA).

Mouse data: The *t*-test (with Welch’s correction) was used if the difference in variances was <4 fold. The *t* test significance and variance were determined using Microsoft Excel. Otherwise, the non-parametric Kolmogorov-Smirnov exact test was performed using GraphPad Prism 10 (GraphPad Software, Boston, MA, USA).

## 3. Results

### 3.1. Omicron Sublineages and Viral Read Counts from COVID-19 ICU Patients

Human nasopharyngeal swab samples were obtained from consented COVID-19 patients admitted to the Intensive Care Unit (ICU) at the Royal Adelaide Hospital (South Australia) between October 2022 and August 2023. All patients had received between one and three doses of antiviral treatment (remdesivir) prior to collection of nasal swabs. The virus sublineages from patients were classified by the diagnostic laboratory at the hospital using RT-PCR of nasopharyngeal swab samples, with viral infections grouped into four omicron sublineages (Figure 1a, x axis).

Nasopharyngeal swab samples from 49 ICU patients were collected into TRIzol^®^ reagent and were transported to QIMR Berghofer. RNA was isolated, with 37 patient samples yielding sufficient RNA (≥100 ng) for library preparation and RNA-Seq (Appendix A). Viral read counts, expressed as viral reads per million (RPM), for this patient cohort ranged across >5 logs. High viral RPM (>5000) were only seen in patients infected with BA.5 and BA.2-like sublineage viruses (Figure 1a). However, viral RPM > 3500 were significantly associated with age ≥ 64 years (see below), arguing that the higher viral RPM are due to patient factors rather than being a feature associated with specific sublineage virus infections. The large range in viral RPM (Figure 1a) is perhaps to be expected given several factors including sampling at different times over the course of the infections, differing effectiveness of anti-viral treatment or anti-viral immunity, and/or different levels of the initial infectious viral inocula.

RNA-Seq of 19/37 patient samples yielded sufficient spike sequence data to confirm the prior RT-PCR-based sublineage identification (Figure 1a, black circle outlines; Appendix A). The variant or sublineage for four samples could not be established by either RNA-Seq (insufficient reads) or by the hospital diagnostic laboratory (Figure 1a, Unknown).

### 3.2. Quantitation by RT-qPCR Correlated Well with Viral Read Counts

The same purified RNA samples that were used for RNA-Seq were also analyzed by quantitative reverse transcription polymerase chain reaction (RT-qPCR) using the standard SARS-CoV-2 Sarbeco-E primers [32]. Viral RPM obtained from RNA-Seq correlated well with Cq values from RT-qPCR (Figure 1b). Viral RNA quantitation by these two methods thus provide highly comparable results. This provides a level of cross validation for viral load assessments using these two techniques for RNA isolated from nasopharyngeal swabs, although cost considerations mean that RNA-Seq is unlikely to replace RT-qPCR in such settings.

### 3.3. Phylogenetic Tree of Omicron Sublineages

Out of 37 human nasopharyngeal swab samples analyzed by RNA-Seq, 19 provided sufficient spike sequence data to allow construction of a phylogenetic tree (Figure 1c; Appendix A). Clear clustering of the sublineages was evident, reflecting the groupings applied in Figure 1a. As might be expected, XBF, a recombinant of BA.5.2 and BA.2.75 [40], grouped with the BA.2-like viruses.

### 3.4. Progression of Omicron Sublineages from October 2022 to August 2023

The number of patients infected with an identified sublineage was tracked month by month from October 2022 to August 2023 and is illustrated as a percentage of each sublineage per month (Figure 1d). A trend in sublineage transitions was observed from BA.5, to BA.2-like, to XBB-like, and finally XBC (Figure 1d), consistent with global trends of omicron sublineage evolution [2,3,4,5,41].

### 3.5. Correlations of Viral Read Counts with Patient Metadata

A range of patient metadata was collected, and potential correlates sought. High viral RPM were significantly associated with patients who were ≥64 years of age (Figure 2a), in agreement with previous reports [42,43]. Corticosteroid and baricitinib (JAK inhibitor) treatment showed no significant effect on viral RPM (Figure 2b). Corticosteroid treatment is well established for reducing mortality [44], although a delay in viral clearance has been reported in some [45], but not other studies [46]. Baricitinib treatment also reduces mortality [47] and has also been reported not to affect viral clearance [48], consistent with the data in Figure 2b.

Whether the patients had received a COVID-19 vaccine did not significantly influence the mean viral RPM; however, only three patients did not receive a vaccine (Appendix A). Vaccination has previously been shown to have no impact on nasopharyngeal swab viral loads in patients infected with omicron lineage viruses [49]. High viral RPM were associated with patients that had one or more comorbidities; however, this did not reach significance for this cohort, with only five patients free of comorbidities (Appendix A). Nevertheless, this is consistent with previous studies showing patients with high viral loads often have comorbidities [50]. The length of stay in ICU or in hospital also did not correlate with viral RPM (Appendix A), again consistent with previous studies [51]. The viral RPM were not significantly different between males and females (Appendix A). Viral loads are generally reported to be slightly higher in females [52]. Overall, these results are generally consistent with, and support, previous studies, arguing that the patient cohort described herein is broadly representative of severe COVID-19 patients and is not demonstrably unique or unusual.

### 3.6. RNA-Seq Analysis of Human Gene Expression

There was sufficient RNA (≥100 ng) in 37/49 samples for RNA-Seq. RNA integrity number (RIN) scores ranged from 2.3 to 7.8 (mean 5.4 ± SD 1.4), illustrating that RNA in the nasopharynx, as might be expected, often suffers from a level of degradation. This is a recognized issue in many clinical samples and can be ameliorated by depletion of ribosomal RNA (used herein), rather than enriching by poly(A) capture, prior to library generation [53,54]. Approximately 30 million reads were obtained for each sample with a mean of 75% aligning to the human reference genome (Appendix A), illustrating that human gene expression data can be obtained from most nasopharyngeal swabs.

DeSeq2 was used to identify differentially expressed genes (DEGs) using pairwise comparisons between samples from patients infected with the different sublineages. This approach provided only a small number of DEGs for each comparison, with BA.5 vs. XBC providing the largest number of DEGs (n = 43 DEGs with gene name annotations) (Appendix A). No cogent pathways were identified by *inter alia* Ingenuity Pathway analysis (IPA) [23,24]. However, the top upregulated gene was Growth and differentiation factor 15 (GDF15) (Appendix A) (a member of the TGF-β superfamily), which in COVID-19, increases with tissue damage [55]. Higher levels of GDF15 mRNA may thus simply reflect the higher viral RPM identified in the nasopharyngeal swabs from some BA.5-infected patients when compared with XBC-infected patients (Figure 1a). Large differences in the levels of virus infection might be expected to have a much greater influence on host gene expression than the more subtle differences that might emerge between infections with different omicron sublineage viruses.

To mitigate against the influence of large viral RPM differences, patient samples with comparable viral RPM were used for human gene expression analysis (Figure 3a, yellow shaded box). The exclusion of samples with high (>1000) and very low (1 and 2.3) viral RPM, left four XBC and four BA.5 samples, with comparable means, standard deviations, and variances (Figure 3a). These eight samples had slightly higher RIN scores (range 3.8–7.8, mean 5.6 ± SD 1.2). DeSeq2 again only identified a small number of DEGs; however, 9/10 of the mRNA species that were down-regulated in XBC samples were associated with cilia (Figure 3b). This analysis argues that more ciliated epithelial cells were infected and disrupted in the nasopharynx of XBC-infected patients than in BA.5-infected patients. The genes were the cilia-associated protein CFAP95 [56] and CFAP65 (expressed in lung [57]), ROPN1L [58], MAPK8IP1 [59], RAB36 (paralog of RAB34) [60], CES1 (highly expressed in ciliated epithelium [61]), LDLRAD1 [62], UBXN10 [63], and UNC119 [64].

Using the data from the same samples (Figure 3a, yellow shading), but using the entire expression matrices (rather than just DEGs), a cellular deconvolution analysis was undertaken to estimate the relative abundance of different cell types. The abundance of ciliated epithelial cells emerged as significantly lower in XBC vs. BA.5 samples (Figure 3c), consistent with the DEGs shown in Figure 3b. Of all the cell types identified by cellular deconvolution, only the abundance of ciliated epithelial cells was significantly different between XBC and BA.5 samples (Figure 3d). Cell-type expression matrices for human lung (lower respiratory tract) were used for this analysis, as cell-type expression matrices for cells from the human nasopharynx (upper respiratory tract) are not available. Some nasopharyngeal cell types not found in lungs, or cell types with distinct expression profiles in lungs, may thus not have been identified by this analysis. Nevertheless, identification of ciliated epithelial cells (Figure 3c) might be considered reliable as the structure and function of these cells is broadly similar in the upper and lower respiratory tracts, with ciliated epithelial cells in both locations expressing cilia and cilia-associated proteins that are generally well conserved [17,65,66].

### 3.7. Mouse Nasal Turbinate Infections with Omicron BA.5 vs. XBB Virus Isolates

To obtain further evidence that later omicron sublineage viruses target ciliated epithelium more efficiently in the nasopharynx than earlier omicron sublineage viruses, two omicron isolates were compared. Specifically, a BA.5 isolate (SARS-CoV-2_QIMR03_), obtained in 2022, and a XBB isolate (SARS-CoV-2_UQ01_), obtained in 2023. Both viruses were isolated in Brisbane, Australia from nasopharyngeal swabs donated by non-hospitalized COVID-19 patients and were described previously [22,23,24]. An XBC isolate was not available. Growth kinetics in a range of cell lines illustrated that the BA.5 and XBB virus isolates displayed no significant differences in their ability to replicate *in vitro* (Figure 4a). Omicron viruses can utilize mouse angiotensin-converting enzyme 2 (mACE2), as well as human angiotensin-converting enzyme 2 (hACE2), as receptors. This ability is illustrated herein by replication of BA.5 and XBB in HEK293T-mACE2 and HEK293T-hACE2 cells (Figure 4a).

A series of mouse strains were infected with 5 × 10^4^ CCID_50_ of BA.5 or XBB via the intranasal route and nasal turbinate titers determined at the indicated times. In all mice strains, for either gender and at one or more time points after infection, nasal turbinate titers were significantly higher in XBB-infected mice than in BA.5-infected mice (Figure 4b, red arrows). The mouse strains tested were C57BL/6J mice (that—like all mice—express mACE2), K18-hACE2 mice (that express both hACE2 and mACE2), *Ifnar*-/- mice (that are defective in type I interferon responses), and *Rag1*-/- mice (that are defective in adoptive immune responses) (Figure 4b). Lung titers in the same individual mice were not significantly different (Appendix A), so in the mouse models, the differences between BA.5 and XBB manifested in the upper, but not the lower, respiratory tracts.

For C57BL/6J mice, the experiment shown in Figure 4b was repeated with RNA-Seq used to determine viral RPM (as in Figure 1a). Viral RPM were significantly higher for XBB than BA.5 (Appendix A), consistent with the viral titer data in Figure 4b.

### 3.8. Immunohistochemistry of Murine Nasal Turbinates Infected with BA.5 vs. XBB

C57BL/6J and *Ifnar*-/- mice were infected with BA.5 and XBB as in Figure 4b, and whole heads were fixed and embedded in paraffin. For orientation, an example of a whole head section stained by H&E is shown in Appendix A. To show virus infected cells, such sections were stained by immunohistochemistry (IHC) using an anti-SARS-CoV-2 spike protein monoclonal antibody [39]. For C57BL/6J mice, staining was much less abundant after BA.5 infection (Figure 5a) than after XBB infection (Figure 5b), consistent with the viral titration data shown in Figure 4b. In the nasal turbinates, epithelial cells of the respiratory epithelium, olfactory epithelium, and Malpighian epithelium stained positive (Figure 5a,b). An enlargement of the image showing cilia staining (Figure 5b) is shown in Appendix A, and clearly illustrates the staining of ciliated epithelial cells and their cilia, which protruding into and across the mucus layer [67]. IHC of lung tissue is shown in Appendix A, illustrating similar levels of staining for both viruses, consistent with the viral titration data in Appendix A.

Staining in the nasal turbinates was also much less abundant for BA.5-infected *Ifnar*-/- mice (Figure 5c) when compared with XBB-infected *Ifnar*-/- mice (Figure 5d), consistent with the viral titration data for this strain of mouse shown in Figure 4b.

## 4. Discussion

We present data from a cohort of COVID-19 ICU patients and from mouse model studies that support the view that the omicron sublineages have continued to evolve to increase targeting of ciliated epithelial cells in the nasopharynx. The patient data used RNA-Seq of nasopharyngeal swabs to compare BA.5 with XBC, and the mouse data compared nasal turbinate infection with a BA.5 isolate vs. a XBB isolate. Two independent lines of evidence are therefore provided comparing the earlier BA.5 sublineage with the later XBB and XBC sublineages. Targeting ciliated epithelial cells confers a number of replication advantages including the ability to infect cells by transiting across the mucosal layer via the cilia [13,67], the promotion of infection and spread by disruption of mucocilary clearance [17,18], as well as promoting viral egress and thus transmission through microvillar remodeling [13]. More efficient person-to-person transmission of virus is arguably the primary driver of evolution for SARS-CoV2, including omicron sublineage viruses [1].

Increased targeting of the upper respiratory track by omicron viruses has been associated in COVID-19 patients with pharyngitis [68] and laryngitis [17,69]. However, elevated severity of pharyngitis/laryngitis, as new omicron sublineages emerge, may be difficult to observe as increasing immunity (due to past infections and/or vaccinations) will contribute to reduced infection and disease.

Although we show herein that human gene expression data can be readily obtained from nasopharyngeal swabs using RNA-Seq, the large range in viral RPM complicated the ability to obtain cogent information. Only when a subset, with comparable viral RPM, was analyzed did the signatures for ciliated epithelial cells emerge. Correlations between viral RPM and patient metadata were broadly similar to those published previously, supporting the view that our cohort is not unique or unusual. However, it is difficult to exclude the possibility that certain patient factors somehow contributed to the increased ciliated epithelial cell signatures for XBB vs. BA.5. The metadata for the eight patients (Figure 3a) is provided (Appendix A). It is also possible that these latter signatures reflect the activities of specific virus isolates, rather than being representative of trends within the omicron sublineages in general. We only analyzed a single nasopharyngeal swab for each patient, rather than taking multiple samples from each patient during their stay in hospital; more samples would increase the robustness of the bioinformatic analyses. Larger patient numbers from several cohorts would also clearly help to confirm the current findings, although the relatively high cost of RNA-Seq might limit such endeavors.

The murine experiments provided robust viral titer, IHC and RNA-Seq data arguing for increased nasopharyngeal infections by a XBB vs. a BA.5 isolate. These isolates were obtained from patients unrelated to the ICU cohort described in Figure 1, Figure 2 and Figure 3, and the experiments focused on early time points post infection. The increased nasal turbinate infections were maintained in *Ifnar*-/- and *Rag1*-/- mice illustrating that the increased infections by XBB were not the result of adaptations associated with increased viral suppression of type I interferon activity [70] or adaptive immunity [71], respectively. Again, we cannot exclude the possibility that these results are due to the characteristics of these virus isolates. However, the lack of titer differences between XBB and BA.5 in lungs does provide a direct internal control for these experiments. The very different nasal turbinate XBB and BA.5 titers are clearly not simply due to different abilities of these isolates to infect mouse respiratory epithelial cells (Appendix A).

The reason why XBB replicated better than BA.5 in the nasal turbinates, but not the lungs, is unclear. This stark difference may indicate some subtle differences between ciliated epithelial cells in the upper and lower respiratory tract, with XBB exploiting features specific to the former. Alternatively, XXB’s adaptation to better infect ciliated epithelial cells in the upper respiratory tract is somehow limited to this setting, with the overall architecture, surrounding cell composition and immune environment quite different in the lower respiratory tract [72,73]. However, mouse models do not recapitulate the migration of infection from the upper to the lower respiratory tract seen in humans. Instead, the infection of mice requires the delivery (via the nose) of a viral inoculum directly into the mouse lungs. As mice are obligate nasal breathers, they would rapidly clear the bulk of the inoculum from the nasal passage. Thereafter, mucocilary clearance would likely also be active in clearing more of the viral inoculum [43]. Clearance from lungs might be impeded by the relatively large volume of the viral inoculum occluding the airways, with the mucus barrier and mucocilary clearance also likely to be less effective in this setting. The viral inoculum may thus make a transitory pass across the nasal epithelium, whereas exposure to the viral inoculum may be considerably longer in the lungs. The lack of differences seen in the lungs between BA.5 and XBB (Appendix A) may thus be a feature the mouse infection model. Either way the lung data provide a compelling internal control.

The molecular changes that underpin the increased targeting of ciliated epithelial cells in the nasopharynx have not been addressed by this study. Multiple mechanisms (see above) and multiple viral proteins may be involved [13,17]. RNA-Seq data from nasopharyngeal tissues are unlikely to be helpful for unraveling the mechanisms involved, as *inter alia* such data would likely be dominated by simple loss of ciliated epithelial cell mRNA. Future work to unravel the molecular mechanisms might involve evaluating a range of isolates in the new sophisticated *in vitro* model organoid systems of nasal epithelial that include ciliated epithelial cells and recapitulate the mucus barrier [67]. Single cell transcriptomics might also be applied to such systems, where infection levels and times post infection can be well controlled.

## 5. Conclusions

RNA-Seq of human nasopharyngeal swabs and a series of mouse model studies provided evidence that the later omicron sublineages XBC and XBB target ciliated epithelial cells in the upper respiratory tract more effectively than the earlier BA.5 sublineage. The data suggest that the evolutionary trend (of increased infection of ciliated epithelial cells by omicron viruses in the upper respiratory tract [13,14]) continued as the omicron sublineage evolved from BA.5 to XBB and XBC.

## Figures and Tables

**Figure 1 viruses-17-01631-f001:**
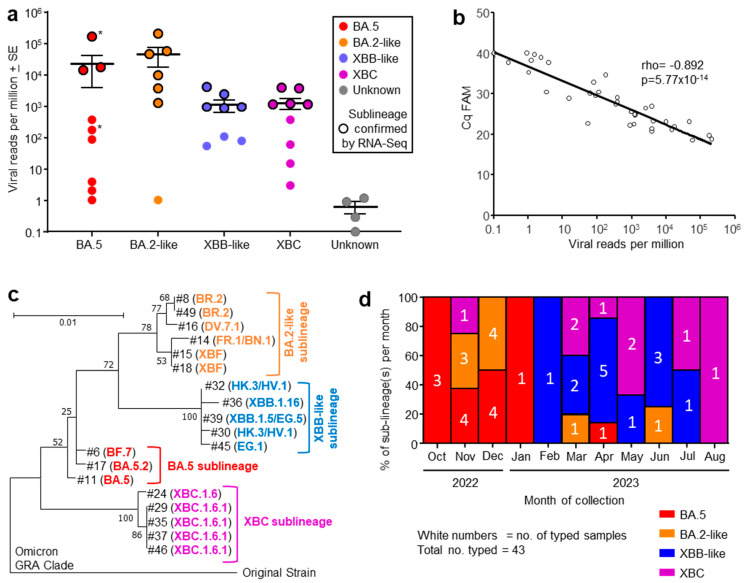
(**a**) Viral reads per million obtained from RNA-Seq for indicated omicron sublineages, with four samples having insufficient reads to allow classification (Unknown). Grouping was based on phylogeny (see (**c**)). The sublineage was determined by RT-PCR at the Royal Adelaide Hospital. Samples for which RNA-Seq confirmation of the sublineage was obtained are indicated by black outline. Patients who died during their hospital stay (*). (**b**) Correlation between RT-qPCR (expressed as Cq FAM) and viral reads per million obtained from RNA-Seq, for the same nasopharyngeal swab samples. Spearman’s correlation *rho* and *p* are provided. (**c**) Phylogenetic tree for SARS-CoV-2 spike protein sequences obtained from RNA-Seq data (n = 19; #—refers to sample numbers in Appendix A). Branch lengths indicate the number of substitutions per site. The tree is rooted using sequence from an original ancestral strain isolate (GenBank ID: MW772455.1). GISAID was used for sublineage classifications; see Appendix A for GISAID annotations and amino acid sequences. (**d**) Percentage of each sublineage collected for each month from October 2022 to August 2023. The number of samples collected for each sublineage collected for each month are shown in white text.

**Figure 2 viruses-17-01631-f002:**
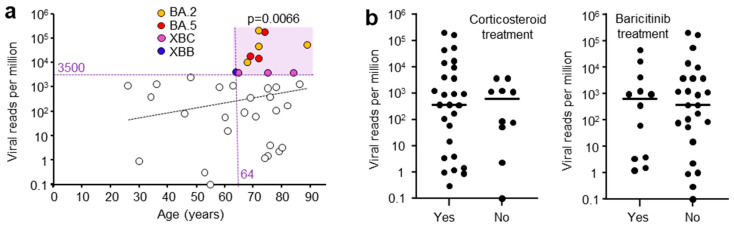
(**a**) Viral reads per million (RPM) and patient age. Viral RPM of >3500 RPM were significantly associated with patients that were ≥64 years of age (light purple shading). Statistics by Fischer’s exact test (*p* = 0.0066). (Dashed trend line did not reach significant). (**b**) Viral RPM for patients that received (Yes) or did not receive (No) corticosteroid or baricitinib (JAK inhibitor) treatment. Horizontal bars represent mean viral RPM. No significant differences were evident for Yes vs. No (*p* = 0.54 Corticosteroid, *p* = 0.98 Baricitinib, Kolmogorov-Smirnov tests).

**Figure 3 viruses-17-01631-f003:**
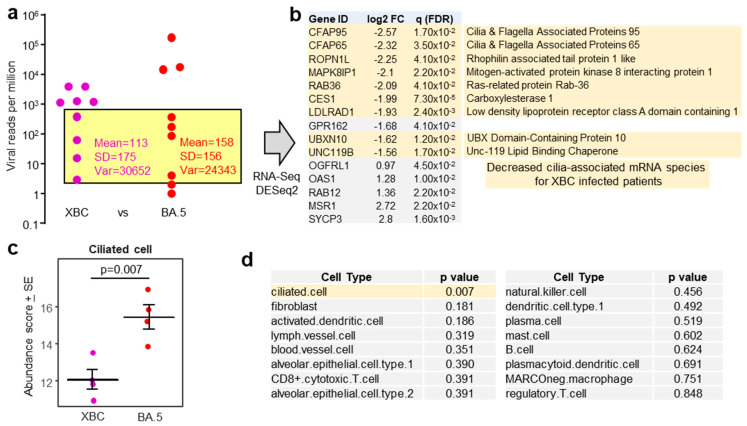
(**a**) Viral reads per million (RPM) are presented as in Figure 1a, but with XBC placed to the left of BA.5. The samples in the yellow shaded box provided a XBC n = 4 vs.BA.5 n = 4 comparison for nasopharyngeal swabs with comparable viral RPM, excluding samples with high (>1000) and very low (1 and 2.3) viral RPM. (**b**) For XBC vs. BA.5, most of the mRNA species (DEGs) significantly down-regulated in XBC samples (i.e., negative log2 fold change relative to BA.5) were genes associated with cilia (pale yellow shading). (**c**) Relative abundance of ciliated cells as determined by cellular deconvolution using human lung cell atlas. Statistics by *t* test. (**d**) A full dataset of cell types identified by cellular deconvolution, showing the cell types with significant (pale yellow shading) or non-significant (grey shading) differences in relative abundance between XBC and BA.5 samples. Statistics by *t* tests.

**Figure 4 viruses-17-01631-f004:**
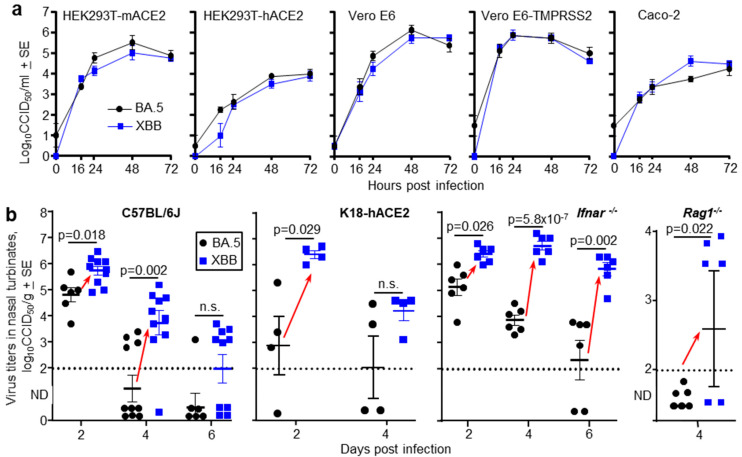
(**a**) Growth kinetics of BA.5 and XBB in five different cell lines *in vitro*. The data represent the mean viral titers from four replicate wells (n = 4) for each virus and cell line. Supernatants from infected cell lines (MOI = 0.1) where harvested at the indicated times and titers determined by CCID_50_ assays. (**b**) The indicated mouse strains were infected with of BA.5 or XBB via the intranasal route (dose 5 × 10^4^ CCID_50_ for each virus). C57BL/6J, *Ifnar*-/- and *Rag1*-/- mice were females and K18-hACE2 mice were males. At the indicated time mice were euthanized and nasal turbinate surgically removed and virus titers in the tissue determined by CCID_50_ assays. Limit of detection ≈ 2 log_10_CCID_50_/g. ND—Not Detected (below limit of detection). Each data point represents one mouse. Red arrows indicate where nasal turbinate virus titers in XBB infected mice are significantly higher than those in BA.5 infected mice. Statistics by *t* tests or Kolmogorov Smirnov exact tests, with Kruskal Wallis test used for *Rag1*-/- mice; values below the level of detection were set to zero. n.s.—not significant.

**Figure 5 viruses-17-01631-f005:**
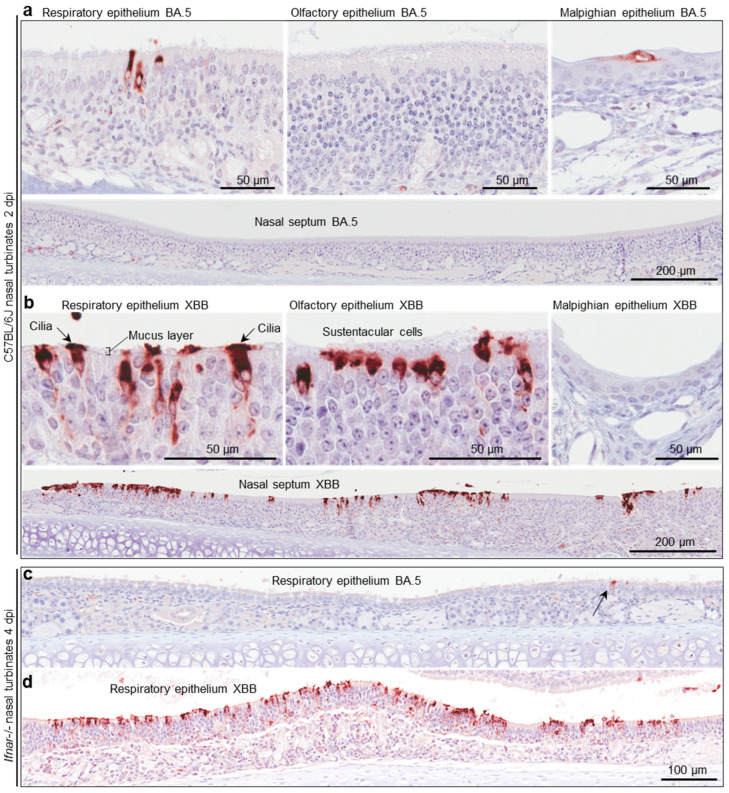
Immunohistochemistry of murine nasal turbinates with anti-SARS-CoV-2 spike antibody (**a**) C57BL/6J mice were infected with BA.5 (as described in Figure 4b) and at two days post infection (2 dpi) were euthanized, and whole heads fixed, decalcified, and embedded in paraffin. Sections were stained with an anti-SARS-CoV-2 spike monoclonal antibody. Several regions of the nasal turbinates are shown. (**b**) As for a, but infection with XBB. (**c**) IHC of respiratory epithelium in the nasal turbinates from *Ifnar*-/- mice infected with BA.5 (arrow indicates an infected cell). (**d**) As for c but after infection with XBB.

## Data Availability

The original contributions presented in this study are included in the article/Appendix A. Further inquiries can be directed to the corresponding authors, and the RNA-Seq fastq files are available from NCBI SRA BioProject ID: PRJNA1313398.

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
