# Peer review of "Evidence from COVID-19 Patients and Murine Studies for a Continuing Trend Towards Targeting of Nasopharyngeal Ciliated Epithelial Cells by SARS-CoV-2 Omicron Sublineages"

_viruses, 2025, doi:10.3390/v17121631_

Round 1
Reviewer 1 Report
Comments and Suggestions for Authors
Carolin et all compared the SARS-CoV-2 infection differences in omicron lineages by RNAseq profiling of nasopharyngeal swab. It was known that omicron variant of the virus attacks upper respiratory tract more efficiently than lower respiratory tract infected by earlier strains. They indeed observed that the increased trend of infection in ciliated epithelial cells in upper respiratory tract in later omicron variants. They validated their results by infecting earlier (BA.5) and later (XBB) omicron strain in murine model.
Results are presented well with conclusion. However, some concerns need to be fixed.
- Line 230, the accession number of original strain should be mentioned.
- Figure 2b, corticosteroid treatment appears to be increased viral loads. Although, it did not reach significance, author could mention the p value in the text (in line 260) and a sentence for explanation supporting ref 43.
- In figure 4b authors presented results of nasal turbinate titer but did not describe the measurement and its criteria in the methods.
- In discussion, authors should mention the observed clinical significance (differences) of excess use of ciliated epithelial cells by later variants of omicron. Because later omicron strains are less virulent than former strains. Again, ciliated epithelial cells are present in many parts of the body, such as brain ventricles, testis, ovary etc.
Reviewer 2 Report
Comments and Suggestions for Authors
The manuscript by Carolin and colleagues examines whether Omicron sublineages evolve towards targeting nasopharyngeal ciliated epithelial cells. Using nasopharyngeal swabs from 37 patients and RNA-seq analysis, the authors identified Omicron sublineages in 33 patients from 43 samples. I have several concerns regarding the methodology used in the manuscript. First, the authors do not discuss the differences in the Spike protein from these sublineages, which would be informative to the reader (and the reviewer). Second, when investigators analyze the evolution of viruses (in this case, the sublineage) in a patient (or group of patients), generally the sublineage in a patient is established at time 0 and then compared with sequential analyses at different time points (e.g., 1 month, 2 months, up to 10 months). This does not appear to have been done, as confirmed by the authors' subsequent statement: "Nasopharyngeal swabs were collected from 49 COVID-19 patients admitted to the ICU of the Royal Adelaide Hospital from October 2022 to August 2023." Here, a series of samples was collected at a specific time point, with no sequential samples collected and analyzed. Third, the clinical parameters of these patients are not discussed. For example, what was the approximate time post-infection when the samples were collected, and did these patients go on to develop severe infection and/or succumb to the infection? This data would be important for such a study. Other comments:
- Lines 22 and 84: The authors state that swabs were taken from 37 (line 22) and 49 (line 84) patients. Is it 37 or 49? Please correct.
- Line 72: Please change “tissue titrations” to “virus titrations.”
- Line 109: Please define “RIN.”
- Line 183: The authors state in this sentence, “times and tissue titers.” However, tissues are not titered; the level of infectious virus in tissues is titered. The authors should change this to, “times and virus titers in tissues were…”
- Lines 211-212: In the sentence with “higher viral RPM only seen in patients infected with BA.5 and BA.2-like sublineage viruses.” Please clarify the significance of this phrase.
- Figure 1d: In this figure, the number of samples analyzed and the sublineage presented. The authors state, “A clear progression was evident from BA.5, to BA.2-like, to XBB-like, and finally XBC (Fig. 1d).” It is unlikely that the numbers presented are statistically significant (1-XBB-like and 2-XBC). I suggest that the authors rephrase this sentence as, “A trend was observed from BA.5, to BA.2-like, to XBB-like, and finally XBC (Fig. 1d).
- Lines 298-301: The sentences, "Although not significant, the mean viral RPM was ≈17 times higher in BA.5 compared with XBC (Fig. 1a); more tissue damage might thus be envisioned in the BA.5-infected cohort. Large differences in the levels of virus infection might be expected to have a much greater influence on host gene expression than the more subtle differences that might emerge between infections with different omicron sublineage viruses." should be removed. This statement is based on viral RPM as the readout for viral loads, which are two different parameters that cannot be compared (discussed above in comment 7. Second, to make the above statement, the authors should quantify the levels of virus recovered from the nasopharynx.
- Line 357: "tissue titers" should be changed to "virus titers."
- Figure 5: Figure 5a shows immunocytochemistry of C57BL/6J inoculated with BA.5 virus and XBB virus. Sections were stained with an antibody against the Spike protein. Shown are micrographs of the respiratory epithelium, olfactory epithelium, and Malpighian epithelium in a BA.5 inoculated mouse and the nasal septum, respiratory epithelium, olfactory epithelium, and Malpighian epithelium in an XBB inoculated mouse. The authors should include a micrograph of the nasal septum from a BA. 5-inoculated C57BL/6J mouse to provide a better comparison.
- Supplemental Figure 1. In this figure, the authors need to understand the difference between "viral loads" and "viral reads per million." Viral loads generally refer to cell-free virus, usually isolated from plasma. I am not saying there is no correlation between viral load and viral reads per million (RPM), but the two parameters are not the same and are not interchangeable. Also, the authors use the term "viral reads per million", which needs clarification. What does "million" refer to? It is not stated in the manuscript. Is it "cells" or "total reads"?
- Supplementary Figure 2. In the Figure legend, 'titer' is written as 'titre'. Please be consistent. Also, the authors show infection of the bronchial epithelium by both BA.5 and XBB at 2 days post-infection. Did either of these viruses infect cells within the alveoli? A sentence in the results would suffice.
Reviewer 3 Report
Comments and Suggestions for Authors
My comments are as follows: This is a well-designed and timely study that addresses the evolving cell tropism of SARS-CoV-2 Omicron sublineages. The central hypothesis—that later Omicron sublineages (XBB, XBC) continue the trend of increased targeting of upper respiratory tract ciliated epithelial cells compared to earlier sublineages (BA.5)—is strongly supported by a compelling combination of clinical RNA-Seq data from ICU patients and mechanistic studies using various murine infection models. The human data suggest increased targeting of ciliated epithelial cells by XBC compared to BA.5, and the murine data clearly demonstrate significantly higher infection levels and ciliated epithelial cell infection by the XBB isolate compared to the BA.5 isolate in nasal turbinates. This work is scientifically sound and has important implications for understanding the changing pathogenicity and transmission dynamics of the virus. I recommend that the paper be accepted after minor revisions to address the points of clarity and discussion listed below.
Major comments:
- The authors note that human gene expression analysis was complicated by the large (>5 log) range in viral reads and appropriately chose to compare only XBC and BA.5 samples that had comparable viral read counts for the differentially expressed genes (DEGs) and cellular deconvolution analysis. To ensure the robustness and reproducibility of this critical finding from the human cohort, it is essential that the authors clearly define the subset of samples used for analysis.
- Please explicitly state the final number of BA.5 and XBC samples used for the comparative DEG and cellular deconvolution analysis. More importantly, justify (perhaps as a new supplementary figure or table) that these selected samples were comparable for other confounding variables in ICU patients, such as time since symptom onset, antiviral treatment status, or relevant patient age/comorbidity data, to minimize bias beyond just viral read count.
- The Discussion mentions a striking result from the murine studies: XBB replicated better than BA.5 in the nasal turbinates but showed no titer difference in the lungs of C57BL/6J mice. While the current discussion attributes this to differences in viral inoculum clearance mechanics (i.e., obligate nasal breathing and inoculum volume), a more in-depth discussion on potential biological/molecular mechanisms is warranted. Please expand the discussion section to hypothesize about potential subtle biological differences between ciliated epithelial cells in the upper versus lower respiratory tract that XBB might be exploiting, as suggested in the text. This would strengthen the paper's contribution beyond a purely descriptive finding.
- In Figure 1a, the patient samples are broadly grouped into four sublineages: BA.5, BA.2-like, XBB-like, and XBC. However, Figure 1c uses specific Pango lineages (e.g., BR.2, DV.7.1, EG.5) for the phylogenetic tree. For maximum transparency, please clearly define the specific Pango lineages that were grouped into the "BA.2-like" and "XBB-like" categories, either in the Figure 1 legend or in a revised Table S1. This will help readers understand the clinical and epidemiological context of the groupings.
Minors:
- For your Methed section, the authors note that the K18-hACE2 mice had a wide age range (7–22 weeks old), but were selected so the mean age was similar in each group. In contrast, C57BL/6J and Rag1-/- mice had a tighter age range (7–8 weeks old). Please add a brief justification for the use of the wide age range in the K18-hACE2 group, even with the mean age matched, and for the use of different sexes across the strains (e.g., K18-hACE2 males vs. C57BL/6J females).
- In your conclusion section, the conclusion reiterates the known fact that Omicron generally shows increased infection of ciliated epithelial cells. The final sentence: "The data presented herein suggest that this evolutionary trend continued as the omicron sub-lineage evolved from BA.5 to XBB and XBC" is sufficient and powerful. Consider streamlining the paragraph (Lines 475-478) to focus purely on the new finding of the sublineage evolution, referencing the prior knowledge (Lines 475-476) only as context.
Round 2
Reviewer 2 Report
Comments and Suggestions for Authors
In the revised manuscript, the authors have addressed most of the concerns raised in the previous review. However, the remaining concern about using a single sample to track the virus's evolution remains. The authors state that obtaining multiple nasal samples from these patients would be problematic, a point this reviewer understands. As a compromise, I suggest that the authors clearly state that they are investigating the phenotypic properties of SARS-CoV-2 strains in this population of infected individuals and comparing early infection across different mouse strains.
Author Response
In the revised manuscript, the authors have addressed most of the concerns raised in the previous review. However, the remaining concern about using a single sample to track the virus's evolution remains. The authors state that obtaining multiple nasal samples from these patients would be problematic, a point this reviewer understands. As a compromise, I suggest that the authors clearly state that they are investigating the phenotypic properties of SARS-CoV-2 strains in this population of infected individuals and comparing early infection across different mouse strains.
The concern is a little unclear. Clearly the evolution of the sublineages occurs across a large number of passages, given that transmission (person to person) is the primary selection pressure for the emergence of the sublineages. We have added “More efficient person-to-person transmission of virus is arguably the primary driver of evolution for SARS-CoV2, including omicron sublineage viruses [1].“ to clarify this point.
How the virus might evolve within each patient is likely a distinct issue and may have more to do with immune escape (e.g. Nat Commun 12, 6405 (2021). Nevertheless, we have added “We have also only analyzed a single nasopharyngeal swab for each patient, rather than taking multiple samples from each patient during their stay in hospital; more samples would clearly increase the robustness of the bioinformatic analyses.” to the Discussion.
To clarify the murine XBB vs. BA.5 experiments, we have also added to the Discussion “These isolates were obtained from patients unrelated to the ICU cohort described in Figs. 1-3, and the experiments focused on early time points post infection.”
Reviewer 3 Report
Comments and Suggestions for Authors
Well addressed.
Author Response
We greatly appreciate your feedback.